# Deficiency in Inactive Rhomboid Protein2 (iRhom2) Alleviates Alcoholic Liver Fibrosis by Suppressing Inflammation and Oxidative Stress

**DOI:** 10.3390/ijms23147701

**Published:** 2022-07-12

**Authors:** Yangwenshu Liu, Qin Kuang, Xianling Dai, Minxia Zhan, Li Zhou, Liancai Zhu, Bochu Wang

**Affiliations:** 1Key Laboratory of Biorheological Science and Technology, Ministry of Education, College of Bioengineering, Chongqing University, Chongqing 400030, China; 201919021126@cqu.edu.cn (Y.L.); 202019131137@cqu.edu.cn (Q.K.); 202019131139@cqu.edu.cn (X.D.); zhanminxia@163.com (M.Z.); zhouli@cqu.edu.cn (L.Z.); 2Modern Life Science Experiment Teaching Center, College of Bioengineering, Chongqing University, Chongqing 400030, China

**Keywords:** alcoholic liver fibrosis, inflammation, iRhom2/TACE/NF-κB signaling pathway, oxidative stress, JNK

## Abstract

Chronic alcohol exposure can lead to liver pathology relating to inflammation and oxidative stress, which are two of the major factors in the incidence of liver fibrosis and even liver cancer. The underlying molecular mechanisms regarding hepatic lesions associated with alcohol are not fully understood. Considering that the recently identified iRhom2 is a key pathogenic mediator of inflammation, we performed in vitro and in vivo experiments to explore its regulatory role in alcohol-induced liver fibrosis. We found that iRhom2 knockout significantly inhibited alcohol-induced inflammatory responses in vitro, including elevated expressions of inflammatory cytokines (IL-1β, IL-6, IL-18, and TNF-α) and genes associated with inflammatory signaling pathways, such as TACE (tumor necrosis factor-alpha converting enzyme), TNFR1 (tumor necrosis factor receptor 1), and TNFR2, as well as the activation of NF-κB. The in vivo results confirmed that long-term alcohol exposure leads to hepatocyte damage and fibrous accumulation. In this pathological process, the expression of iRhom2 is promoted to activate the TACE/NF-κB signaling pathway, leading to inflammatory responses. Furthermore, the deletion of iRhom2 blocks the TACE/NF-κB signaling pathway and reduces liver damage and fibrosis caused by alcohol. Additionally, the activation of the JNK/Nrf2/HO-1 signaling pathway caused by alcohol exposure was also noted in vitro and in vivo. In the same way, knockout or deleting iRhom2 blocked the JNK/Nrf2/HO-1 signaling pathway to regulate the oxidative stress. Therefore, we contend that iRhom2 is a key regulator that promotes inflammatory responses and regulates oxidative stress in alcoholic liver fibrosis lesions. We posit that iRhom2 is potentially a new therapeutic target for alcoholic liver fibrosis.

## 1. Introduction

As one of the most important metabolic organs in the human body, the liver is a key hub for the occurrence and development of many normal physiological activities [1,2]. Despite the rapid advancement of human quality of life, the global mortality rate of liver disease is still on the rise. According to one survey, in 2015, the number of deaths caused by liver disease in the Asia–Pacific region accounted for more than half (62.6%) of the global deaths caused by liver disease, and the number of deaths caused by alcohol consumption accounted for 20.8% [3]. After entering the human body, 10–20% of the alcohol is absorbed in the stomach and 75–80% is absorbed in the small intestine. Alcohol enters the liver through the bloodstream and begins to metabolize, and this metabolization causes a kind of injury and is a burden to the liver. The safest amount of alcohol consumption is zero [4]. The average annual consumption of alcoholic beverages per capita in China increased from 4.9 L in 2003–2005 to 7.2 L in 2016, and the average consumption of Chinese drinkers in 2016 was 12.9 L [5]. China’s rapid economic development over the last 30 years, as well as the spread of drinking culture, has caused significant changes in the incidence rate and number of patients with chronic liver disease [6,7]. 

When the liver metabolizes alcohol over a long period of time, it will suffer alcoholic liver disease, leading to fibrosis, cirrhosis, and even cancer. For liver cirrhosis and cancer, there are no effective treatments outside liver transplantation. Unfortunately, the cost of liver transplantation is high, donor organs tend to be in short supply, and survival rates are low [8,9,10]. However, among these symptoms is liver fibrosis, the result of a continuous wound healing response to chronic liver injuries [11] which is reversible [12,13]; thus, antifibrosis therapy is the most effective means of preventing liver disease from progressing to the cirrhosis and cancer stages. That said, the upstream mechanism of liver fibrosis is poorly understood. 

Liver fibrosis is itself a serious threat to human health [14]. It is characterized by the gradual accumulation of extracellular matrix (ECM), which leads to the formation of fibrous scars, thus destroying the normal physiological structure of the liver. Active hepatic stellate cells (HSC) are the main producers of ECM in the process of liver fibrosis [15,16]. Meanwhile, macrophages also play a role in the progression of liver fibrosis. The injury-induced inflammatory response triggers macrophages to enter the liver and produce cytokines and chemokines, inducing HSCs to transform into ECM-producing myofibroblasts [17,18]. Moreover, liver-specific macrophages, namely Kupffer cells (KC), are activated and produce reactive oxygen species (ROS), thereby promoting the activation and migration of HSCs. When HSCs are activated, the liver begins to express α-smooth muscle protein (α-SMA) [19,20]. Oxidative stress and related cell damage promote inflammation, and the increase in proinflammatory cytokines, such as tumor necrosis factor α (TNF-α), interleukin-6 (IL-6), IL-18, etc., further aggravate inflammation [21,22,23,24]. Therefore, oxidative stress and inflammation are important factors in promoting the occurrence and development of liver fibrosis. 

Inactive rhomboid protein2 (iRhom2) is an inactive member of the rhomboid intramembrane proteinase family. TACE (tumor necrosis factor-alpha converting enzyme) is a kind of TNF-α invertase; iRhom2 regulates TACE to control the maturation of TNF-α, abscission, and biological activity in vivo [25,26]. TACE is also known as disintegrin and Ametalloprotease 17 (ADAM17), and it is an important regulator of both the TNFα-TNFα receptor signaling pathway and the EGFR ligand-EGFR signaling pathway. A large number of TACE target molecules exist in vivo, and they are involved in the regulation of various physiological and pathological processes [27]. The iRhom2/TACE pathway has been reported to be associated with diabetic kidney injury in rats [28]. Moreover, it has been reported that regulating iRhom2 significantly reduces the content of TACE/TNFR (tumor necrosis factor receptor) in liver tissue, alleviating liver injuries in mice caused by PM2.5 [29], and that knocking out iRhom2 reduces acute liver injury by blocking the inflammatory response induced by PM2.5 [30]. It has also been reported that iRhom2 plays an important role in the progression of PM2.5-induced renal injury through the activation of TACE/TNFR and the IκBα/NF-κB signaling pathway [31]. We hypothesized that iRhom2 may also be involved in the progression of alcohol-induced liver fibrosis. Therefore, in this study, we investigated the role of iRhom2 in alcohol-induced liver fibrosis and its related signaling pathways to provide new strategies for the treatment of alcoholic liver fibrosis.

## 2. Results

### 2.1. Suppression of iRhom2 Negatively Regulates Inflammatory Response in Human Hepatocyte L02 Cells

Different alcohol concentrations and exposure times have different effects on the mRNA levels of iRhom2 and TACE in L02 and HSC-t6 cells (Figure 1). After L02 and HSC-t6 cells were exposed to 0%, 0.5%, 1.0%, 1.5%, and 2.0% gradient alcohol for 4 h, the expression of iRhom2 and TACE mRNA increased with each increase in concentration. After the L02 and HSC-t6 cells were exposed to 2.0% alcohol for 0 h, 1 h, 2 h, 3 h, and 4 h of gradient time, the expression of iRhom2 and TACE mRNA decreased after reaching the maximum value at 1 h. Therefore, the cells were treated with 2.0% alcohol for 1 h in subsequent experiments.

The effects of iRhom2 knockout on the inflammatory response of L02 induced by alcohol are shown in Figure 2. After wild L02 cells were treated with 2.0% alcohol for 1 h, the mRNA expressions of iRhom2, TACE, TNFR1, and TNFR2 increased significantly. Conversely, for iRhom2-knockout L02 cells treated with alcohol, the mRNA expressions of iRhom2, TACE, TNFR1, and TNFR2 did not change significantly compared to the nonalcoholic treatment group. Meanwhile, from the results of a WB (Figure 2B), we also found that the knocking out of iRhom2 could inhibit increases in iRhom2 and phosphate NF-κB induced by alcohol. In the same way, the expression of IL-1β, IL-6, IL-18, and TNF-α mRNA was different in iRhom2-knockout L02 cells and wild L02 cells after exposure to alcohol. These inflammatory factors increased in the control group and the negative control group, but in the iRhom2-knockout group, they remained consistent with the nonalcoholic treatment group. Thus, iRhom2 could play an essential role in the alcohol-induced inflammatory response of L02 cells; in other words, iRhom2 knockout negatively regulates the inflammatory response of L02 cells induced by alcohol. 

### 2.2. iRhom2 Deficiency Alleviated Alcohol-Induced Liver Fibrosis In Vivo by Reducing Inflammatory Infiltration

In this paper, mice were used to model the development of alcoholic liver fibrosis by gavaging them with an “alcohol–pyrazole–corn oil” mixture. Pyrazole is the main inhibitor of ADH. Findings in the literature indicate that pyrazole alone does not cause liver damage when given to mice [32]. Hepatocellular damage requires pyrazole to act in combination with alcohol. Therefore, the induction of alcoholic liver disease in mice requires the administration of both alcohol and pyrazole. The addition of corn oil may increase fat intake in mice and may mimic the human habit of drinking alcohol accompanied by high-fat foods. In one study, a CCL4 control group showed only slight-to-moderate liver steatosis, and no inflammation or collagen deposition was observed [33,34]. Therefore, in this experimental mouse model, alcohol was the dominant factor, and pyrazole combined with corn oil aggravated the liver burden to induce the formation of liver fibrosis [35,36,37].

A histological analysis showed that iRhom2 deficiency can alleviate alcohol-induced liver lesions, including hepatocyte injuries and fibrous accumulation (Figure 3A,B). For wild mice, long-term alcohol exposure led to obvious liver lesions. The hepatocytes were disordered, enlarged, and accompanied by obvious collagen accumulation and inflammasome infiltration. In contrast, in the iRhom2 gene knockout mice, histological changes in the liver induced by alcohol were significantly alleviated. The results of ELISA and Western blotting also showed that the accumulations of α-SMA and COL I induced by alcohol were blocked by knocking out iRhom2 (Figure 3C,D). Meanwhile, we found that increases in ALT, AST, and ALP in serum induced by alcohol were also inhibited (Figure 3E). The above results indicate that iRhom2 gene knockout can alleviate the liver function injury and liver fibrosis induced by alcohol.

IF staining and qPCR analysis showed that TACE levels in the livers of wild mice increased after long-term exposure to alcohol. Interestingly, the TACE content in the livers of iRhom2-knockout mice remained unchanged after alcohol stimulation compared to the nonalcoholic treatment group (Figure 4A). In the serum and liver tissue of wild mice exposed to alcohol, the expression of IL-1β, IL-6, IL-8, TNF-α, TNFR1, and TNFR2 increased. Conversely, the contents of these factors in the serum and liver tissue of iRhom2-knockout mice remained unchanged after alcohol induction (Figure 4B). On the gene level, IL-1β, IL-6, TNF-α, and IL-18 mRNA levels were not elevated in the liver tissue of iRhom2-knockout mice exposed to alcohol. In wild mice chronically exposed to alcohol, IL-1β, IL-6, TNF-α, and IL-18 mRNA expressions increased significantly (Figure 4C). Moreover, after the wild mice were treated with alcohol, we found that the phosphorylation of NF-κB in their liver tissue significantly increased compared to the nonalcoholic treatment group. However, as for the phosphorylation of NF-κB in the liver tissue of iRhom2-deficient mice, there was no significant difference between the alcoholic treatment group and the nonalcoholic treatment group (Figure 4D,E). Finally, we found that long-term alcohol exposure can significantly increase the level and expression of iRhom2 mRNA in wild mouse liver tissues (Figure 4F). Therefore, we speculate that iRhom2 plays an important role in the inflammatory stress of mouse livers stimulated by alcohol. Knocking out iRhom2 can significantly reduce inflammatory factors and block the TACE/p-NF-κB pathway to alleviate liver injury and liver fibrosis induced by alcohol.

### 2.3. Deficiency of iRhom2 in L02 Cells and Mice Reduced Alcohol-Induced Liver Oxidative Stress and Inhibited JNK Activation

In L02 cells, we found that long-term exposure to alcohol led to oxidative stress, and the knockout of iRhom2 inhibited the oxidative stress of L02 cells induced by alcohol. After 1 h of induction with 2% alcohol, the expressions or contents of SOD, CAT, ROS, HO-1, and Nrf2 in iRhom2-knockout L02 cells were consistent with those of cells that were free of alcohol (Figure 5A–C). This suggested that iRhom2 is a key gene in the activation of the oxidative stress response induced by alcohol, which could be related to the Nrf2/HO-1 signaling pathways. 

In wild mice, we also found that long-term exposure to alcohol significantly promoted the level of MDA and OH- in liver tissue. As for iRhom2-knockout mice, the expression of MDA and OH- remained unchanged after alcohol induction. Meanwhile, exposure to alcohol decreased the contents of SOD, GSH, and T-AOC in the livers of wild mice but did not change the contents of these antioxidative factors in the livers of iRhom2-knockout mice (Figure 6A,B). In addition, the expression of HO-1, Nqo1, and Nrf2 remained unchanged by iRhom2 gene knockout after alcohol induction, as compared to the nonalcoholic group. Moreover, after being exposed to alcohol, the overexpression of phosphorylated JNK was significantly reduced in the livers of iRhom2-knockout mice compared to those of wild mice treated with alcohol (Figure 6C–E). Therefore, iRhom2 may play a key role in oxidative stress and JNK activation induced by alcohol.

## 3. Discussion

In this study, for wild mice, long-term alcohol exposure led to liver fibrosis. We found that iRhom2, as a key gene, can regulate TACE, thus activating inflammatory stress and leading to liver fibrosis caused by alcohol. The absence of iRhom2 may play a protective role both in vitro and in vivo. Due to stimulation from alcohol, pathological changes, such as fibrotic protein deposition, appeared in the livers of wild mice. However, H and E and Masson staining the pathological sections showed that there was no fibrotic protein deposition in the liver tissue of iRhom2-knockout mice, and the hepatocytes were arranged in an orderly manner. Both α-SMA and COL I are proteins associated with hepatic scar formations and fibrosis, which are expressed by activated HSC cells differentiated into myofibroblasts [38,39]. In iRhom2-knockout mice, after long-term exposure to alcohol, there was no significant change in α-SMA or COL I protein content. Similarly, ALT and AST are enzymes that promote amino acid and ketoacid amino acid transfer. When the liver is damaged by alcohol stimulation, the permeability of the cell membrane increases, and ALT and AST in hepatocytes are released into the bloodstream, resulting in an increase in ALT and AST levels in serum. Therefore, their abnormal increase is an important and sensitive biochemical marker for liver function evaluation [40,41]. After alcohol stimulation, the ALT and AST content in the serum of iRhom2-knockout mice did not increase but that of wild mice increased significantly. ALP is a hydrolase that demonstrates strong activity in alkaline conditions, and it exists in blood in various forms. Most ALP comes from bone osteoblasts and the liver. Therefore, an increase in ALP in serum also indicates the possibility of liver injury, at least to a certain extent [42,43]. However, in the serum of iRhom2-knockout mice, ALP levels did not rise with the continuous stimulation of alcohol. Furthermore, iRhom2 knockout resulted in a TACE/TNFR/NF-κB lockage of the signaling pathway, leading to the inhibition of inflammatory cytokine activation. TACE is the key enzyme of released TNF-α [44]; it cleaves TNFR, dissociating the TNFR binding complex and terminating the signal transduction regulated by TNFR [45]. In this study, long-term alcohol exposure resulted in liver inflammation, and the contents and mRNA expressions of TNF-α, TNFR1, TNFR2, IL-1β, IL-6, and IL-18 in the liver tissue samples of wild mice increased significantly. In the livers of wild mice exposed to alcohol in the long term, the NF-κB signaling pathway was activated. However, in iRhom2-knockout mice, because iRhom2 specifically inhibits soluble TNF-α [46], the proinflammatory pathway was blocked, and the liver’s inflammatory response was alleviated, possibly because the TACE/TNFRs/NF-κB signaling pathway was inhibited. This led to a reduction in the secretion of proinflammatory cytokines. Similar results were observed in human hepatocytes incubated in 2% alcohol. Knocking out iRhom2 to inactivate TACE/TNFRs/NF-κB pathways inhibits inflammation. Therefore, we hypothesized that knocking out iRhom2 reduces liver fibrosis due to the inhibition of inflammation.

The pathogenesis of alcoholic liver disease is complex and unclear, but the oxidative metabolites of alcohol, such as acetaldehyde and ROS, are key factors in liver pathology. Oxidative stress is caused by an imbalance between ROS and antioxidants [47,48,49]. In this study on cells and mice, we showed that long-term alcohol intervention can lead to ROS production, antioxidant decrease, and Nrf2/HO-1 activation. Conversely, the oxidative stress response of iRhom2-knockout mice did not increase after alcohol stimulation compared to the nonalcoholic treatment group. JNK activation is a key pathway leading to oxidative stress [50,51,52,53]. Moreover, iRhom2 has been previously proven to regulate JNK and induce the oxidative stress response to liver injuries [54]. In this study, JNK was found to be significantly activated by alcohol, and the expression of Nrf2/HO-1 was enhanced in the wild animals and cells. However, with respect to iRhom2-knockout mice and cells, alcohol treatment did not induce the phosphorylation of JNK. Therefore, we believe that knocking out iRhom2 contributes to the suppression of oxidative stress, thus reducing alcoholic liver fibrosis.

For the first time, we have provided evidence that iRhom2 is essential for regulating the progression of liver fibrosis and injuries caused by alcohol exposure, which is most likely to be achieved by activating the TACE/TNFR/NF-κB and JNK/Nrf2/HO-1 signaling pathways (Figure 7). Thus, the knockout of iRhom2 may form the basis for a promising and novel therapeutic strategy for alcoholic liver fibrosis. Further research should be conducted in the future to explore the feasibility of gene therapy using iRhom2 to treat alcoholic liver fibrosis.

## 4. Materials and Methods

### 4.1. Animals and Experimental Design

Male, 6-to-8-week-old wild-type mice were purchased from Beijing Vital River Laboratory Animal Technology Company Ltd. (Beijing, China). The iRhom2-knockout (iRhom2−/−) mice used in this study, based on a C57BL/6 background and weighing 20–25 g, were donated by Professor Tan Jun from the Chongqing University of Education [29]. They were housed in a specific pathogen-free (SPF), temperature- and humidity-controlled environment (25 ± 2 °C, 50 ± 5% humidity) with a standard 12 h light/12 h dark cycle and with food and water in cages. A mixture of 35% ethanol (10 mL kg^−1^ d^−1^), corn oil (2 mL kg^−1^ d^−1^), and pyrazole (25 mg kg^−1^ d^−1^) was fed to them via gavage, once in the morning and once in the evening. From the second week onward, CCl4 corn oil solution (CCl4:corn oil = 1:3) was injected intraperitoneally twice a week at a dose of 0.3 mL/kg until the end of the 15th week (Figure 8). After alcohol exposure for 15 weeks, all mice were sacrificed for blood collection. Liver tissue was isolated from the mice for further study. All animal experiments were approved by the Ethics Committee of the Chongqing University Cancer Hospital.

### 4.2. Cells and Culture

Rat hepatic stellate cells (HSC-t6) were purchased from Shanghai Cell Research Institute (Shanghai, China). Human liver (L02) cells were donated by Professor Liling Tang of Chongqing University. All cells were incubated in DMEM or RPMI 1640 medium (Hyclone, Logan, UT, USA), supplemented with 10% fetal bovine serum (Hyclone) and 100 U/mL penicillin streptomycin at 37 °C in a humidified atmosphere containing 5% CO_2_. The in vitro model of alcohol damage was established thusly: first, the cell viabilities of L02 and HSC-t6, after being treated with different concentrations of alcohol for 4 h, were determined using CCK8 (Appendix A). Second, according to the CCK8 results, 0%, 0.5%, 1.0%, 1.5%, and 2.0% ethanol media were selected to treat cells for 4 h, and expression changes in iRhom2 and TACE were determined with qPCR. Then, a 2.0% ethanol medium was selected to culture cells for 0 h, 1.0 h, 2.0 h, 3.0 h, 4.0 h, and 5.0 h. Expression changes in iRhom2 and TACE were also determined. Finally, L02 cells were cultured in 2.0% ethanol medium for 1 h to establish an in vitro model of alcohol-damaged hepatocytes. 

### 4.3. CRISPR/cas9

The pCAG-hCas9 plasmid was selected, and the human iRhom2 gene sequence was searched in the NCBI Gene Bank. The iRhom2 (gene number NM_001005498.4) sequence was analyzed. Three sgRNA sequences (Appendix A) were designed using the online design site CRISPR Direct (http://crispr.dbcls.jp/, accessed on 16 June 2021) to identify knockout targets, and they were synthesized by Bioengineering (Shanghai, China) Co. The plasmids were digested using BsmB I restriction endonuclease (Thermo Fisher, ER0451, Waltham, MA, USA), FastAP (Thermo Fisher, EF0651), and 100 mM DTT (freshly prepared), then cut for gel recovery (Tkara, 9762, Dalian, China). The synthesized oligonucleotide single-stranded SgRNA was annealed with T4 polynucleotide kinase (Thermo Fisher, EK0031, Waltham, MA, USA) to form a dimer, and the SgRNA dimer was ligated to the enzymatically cleaved plasmid using a fast ligase (NEB, M2200S, Beijing, China). The ligand system was then transformed into DH5-α receptor cells, coated on ampicillin-resistant LB medium plates, and incubated at 37 °C for 12–14 h. Individual colonies were picked and shaken, and the plasmids were extracted from the bacterial broth and sent to Bioengineering (Shanghai, China) Co., Ltd. for sequencing and identification to determine successful ligation (Appendix A).

### 4.4. Plasmids Transfection

Constructs for the iRhom2-knockout plasmid and the empty vector (EV) plasmid were used in a plasmid extraction kit (OMEGA, D2156-00, Guangzhou, China) and were measured at concentrations >90%. All constructs were confirmed with DNA sequencing. The plasmids were transfected into L02 cells using lipofectamine 3000 (Thermo Fisher, L3000015) according to the manufacturer’s instructions, incubated at 37 °C in a humidified atmosphere containing 5% CO_2_ for 24 h, and successful transfection was observed under a fluorescent microscope (Appendix A).

### 4.5. Quantitative Real-Time PCR (qPCR)

Total RNA was isolated using Trizol reagent (Invitrogen, Shanghai, China) in accordance with its instructions. First-strand cDNA was synthesized using Reverse EasyScript One Step gDNA Removal and cDNA Synthesis SuperMix (TAKALA, Dalian, China). The RNA expression levels were determined using SYBR^®^ Green mixture (Qiagen, Shanghai, China) reagent on ABI PRISM 7900HT detection systems (Applied Biosystems, Foster City, CA, USA). The primer sequences are provided in Appendix A. GAPDH was taken as an internal control, and the gene expressions were assessed using the 2^−ΔΔCt^ method.

### 4.6. Western Blot Analysis (WB)

Total protein was extracted from liver tissue samples or cells using RIPA lysis buffer (Solarbio, Beijing, China). Then, the final liquid supernatants were harvested via centrifugation at 13,500 rpm for 30 min. Protein concentrations were calculated using PierceTM Rapid Gold BCA Protein Assay Kit (Thermo Scientific, Waltham, MA, USA). Next, the protein samples were subjected to 10% or 12% SDS-PAGE and transferred into PVDF membranes (Millipore, Bedford, MA, USA). The membranes were blocked in 5% nonfat milk, then incubated with specific primary antibodies at 4 °C overnight. The membranes were incubated with HRP-conjugated secondary antibody. After washing, protein bands were visualized using Super ECL Detection Reagent (Yeasen Biotech Co., Ltd., Shanghai, China) and exposed to Kodak (Eastman Kodak Company, Rochester, NY, USA) X-ray film. Corresponding protein expression was determined as a gray value (ImageJ, Version 1.4.2b, National Institutes of Health, Bethesda, MA, USA), standardized to the housekeeping gene (GAPDH), and expressed as a fold of the control.

### 4.7. Serum Biochemistry

Serum alanine transaminase (ALT), aspartate aminotransferase (AST), and serum alkaline phosphatase (ALP) levels were detected using corresponding kits that were purchased from Nanjing Jiancheng Bioengineering Institute (Nanjing, China). 

### 4.8. Enzyme-Linked Immunosorbent Assay

The following enzyme-linked immunosorbent assay (ELISA) kits were used: α-SMA (JLC2816), COL I (JLC2698), TNF-α (LLC3924), TNFR1 (JLC3924-11), TNFR2 (JLC3924-12), IL-1β (JLC3580), IL-6 (JLC3601), IL-18 (JLC3558), and p-NF-κB (JLC3669). iRhom2 (JLC3010) levels in liver were tested according to the Shanghai Jing Anti Biological Engineering Co’s introductions. All ELISAs were performed according to the manufacturers’ instructions.

### 4.9. Oxidative Stress Analysis In Vivo and In Vitro

Superoxide dismutase (SOD), malondialdehyde (MDA), glutathione (GSH), hydroxyl-free radical (OH-), and total antioxidant capacity (T-AOC) levels were detected using corresponding, commercially available kits (Nanjing Jiancheng Bioengineering Institute). Total ROS generation in cells was also determined using the CellROX Oxidative Stress Reagents.

### 4.10. Immunofluorescence (IF), H and E and Masson Staining

Liver tissue samples from each group of mice were fixed in 10% *v*/*v* formalin/PBS, then embedded in paraffin and sectioned at 4 µm for staining with hematoxylin and eosin (H and E) as well as Masson trichrome staining. Images were obtained using a microscope. All sections were detected by 3 histologists without knowledge of the treatment procedure. IF analysis for phosphorylated JNK (ab131499, 1:200, Abcam, Cambridge, MA, USA), phosphorylated NF-κB (ab86299, 1:200, Abcam, Cambridge, MA, USA), and TACE (ab2051, 1:200, Abcam, HongKong, China) was performed as previously described. In brief, liver tissue sections were incubated in 3% H_2_O_2_ to block endogenous peroxidase activity for 10 min, and 5% bovine serum albumin (BSA, Shanghai Boao Biotechnology Co., Ltd., Shanghai, China) was used to block nonspecific binding for 1 h. Then, tissues were incubated with primary antibodies. 

### 4.11. Data Analysis

Data are expressed as the mean ± standard deviation (SD). Statistical analyses were performed using GraphPad PRISM (version 6.0; GraphPad Software, San Diego, CA, USA) by analysis of variance with Dunnett’s least significant difference post hoc tests. A *p*-value of <0.05 was considered significant.

## Figures and Tables

**Figure 1 ijms-23-07701-f001:**
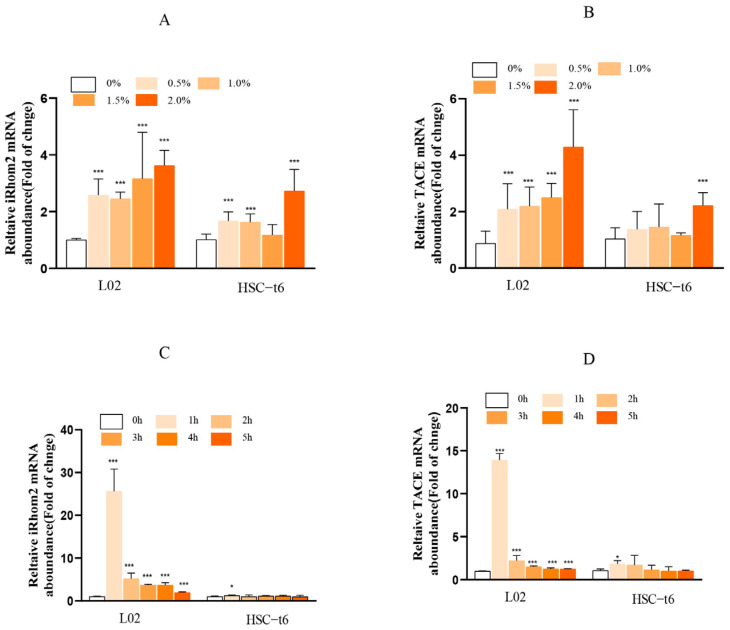
The mRNA expression of iRhom2 and TACE in different cells cultured with alcohol: (**A**) L02; and (**B**) HSC-t6 were treated with alcohol at varying concentrations (0, 0.5, 1.0, 1.5, and 2.0%) for 4 h; (**C**) L02; and (**D**) HSC-t6 were induced with 2.0% alcohol for different time periods (0, 1, 2, 3, 4, and 5 h). The data are expressed as the mean value ± SD (*n* = 6). * *p* < 0.05 and *** *p* < 0.001 vs. the control group (0% or 0 h group).

**Figure 2 ijms-23-07701-f002:**
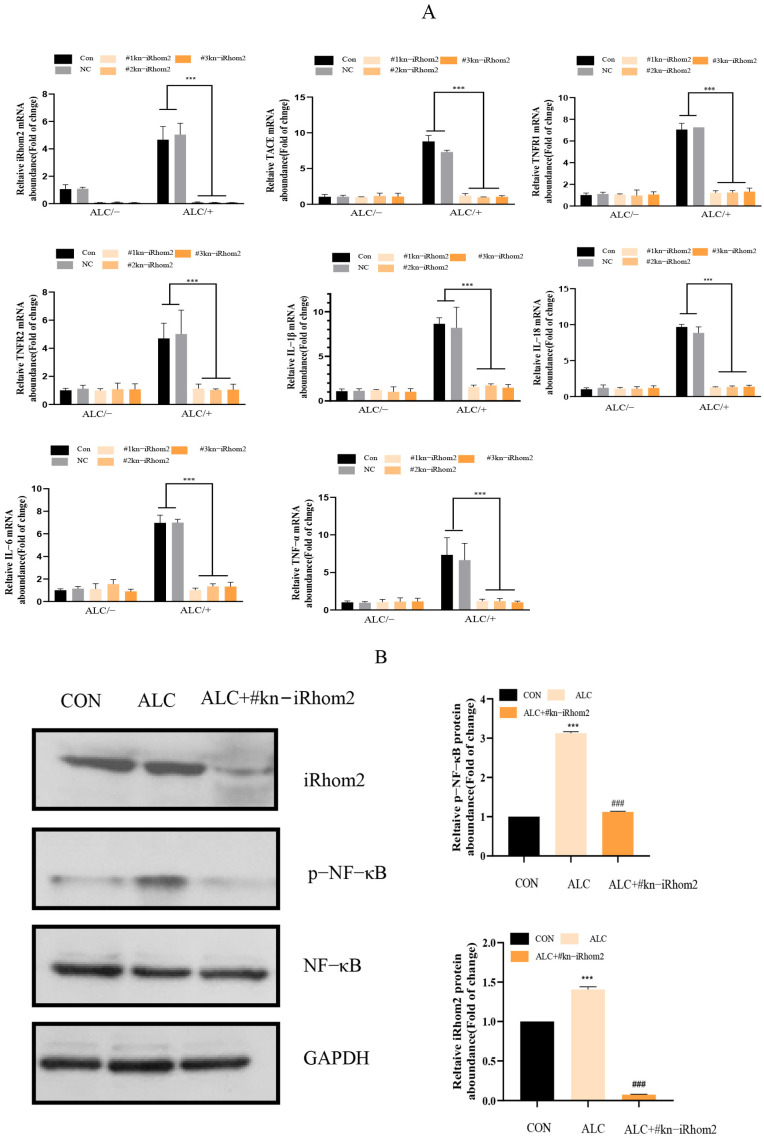
The qPCR showing that the knockout of iRhom2 suppressed the transcriptional expression of iRhom2, TNFR1, 2, IL-1β, IL-6, IL-18, TNF-α, and TACE mRNA in L02 cells exposed to alcohol (**A**); WB showing that the knockout of iRhom2 decreased the expression of iRhom2 and the phosphorylation of NF-κB in L02 cells induced by treatment with alcohol (**B**); The data are expressed as the mean value ± SD (*n* = 6). *** *p* < 0.001 vs. the control group (CON and NC group). ### *p* < 0.001 vs. the ALC group.

**Figure 3 ijms-23-07701-f003:**
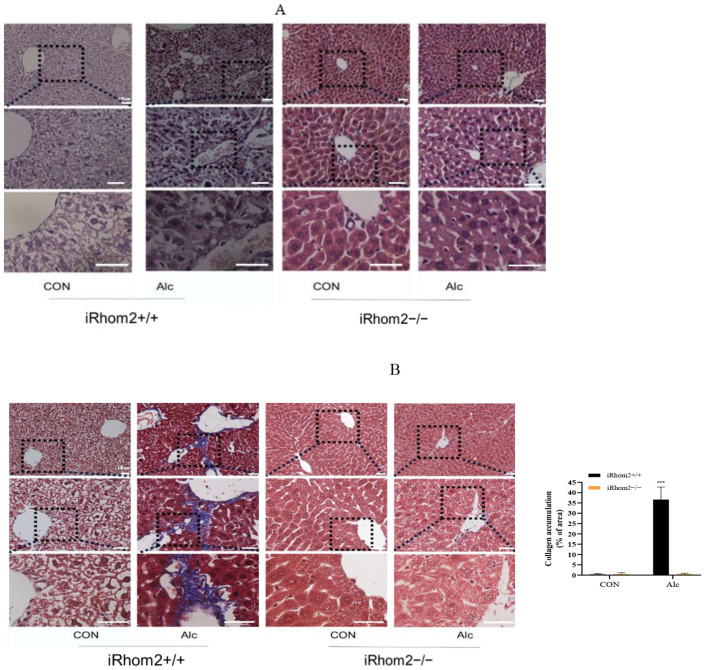
H and E and Masson tissue sections showing that the knockout of iRhom2 can reduce liver injury and fibrotic collagen, and the content of collagen stained by Masson was quantified (**A**,**B**); Elisa and WB showing that the contents of α-SMA and COL I in normal mice exposed to alcohol for a long period increases. However, the content of fibrosis-specific proteins in the livers of iRhom2-knockout mice induced by alcohol remained unchanged (**C**,**D**); The ALT, AST, and ALP levels in mouse serum proved that the knockout of iRhom2 can reduce liver injury (**E**); The data are expressed as the mean value ± SD (*n* = 6). * *p* < 0.05, and *** *p* < 0.001 vs. the iRhom2+/+/CON group.

**Figure 4 ijms-23-07701-f004:**
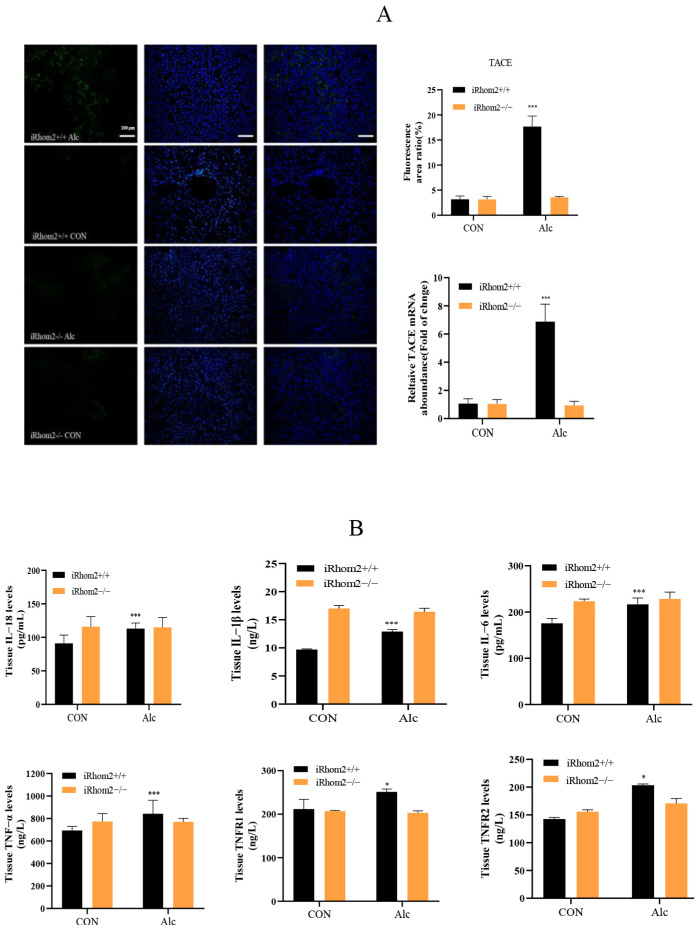
The iRhom2 gene reduces alcohol-induced inflammation: (**A**) TACE levels measured with IF and qPCR; (**B**) the presence of IL-1β, IL-6, IL-18, TNF-α, TNFR1, and TNFR2 in tissue, calculated with ELISA; (**C**) IL-6, IL-18, IL-1β, and TNF-α in liver tissue, calculated with qPCR; (**D**) expression of p-NF-κB in liver tissue, analyzed with IF; (**E**) WB analysis of p-NF-κB in liver tissue; (**F**) iRhom2 levels in mouse livers, analyzed with qPCR and ELISA. The data are expressed as the mean value ± SD (*n* = 6). * *p* < 0.05 and *** *p* < 0.001 vs. the iRhom2+/+/CON group.

**Figure 5 ijms-23-07701-f005:**
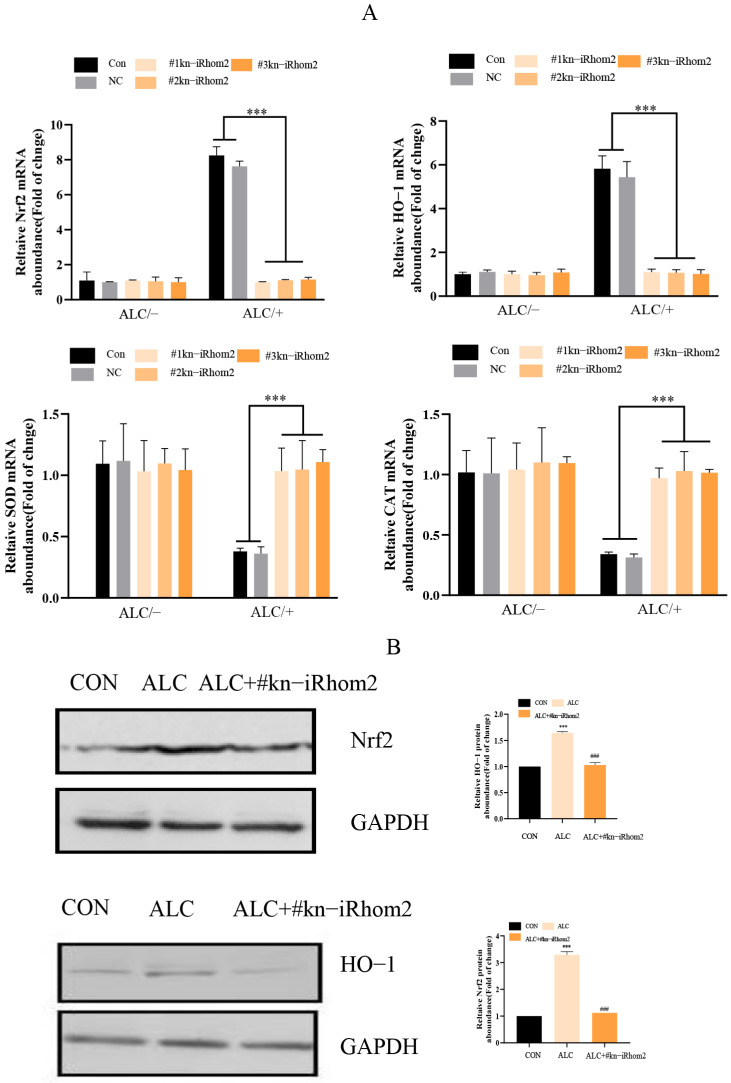
The iRhom2 knockout reduced alcohol-induced oxidative stress and JNK activation in L02 cells; (**A**) The mRNA expression of L02 cells containing a blank vector after the administration of alcohol induction (NC) or the knockout of iRhom2 (kn-iRhom2) was analyzed with the qPCR method to calculate the levels of cellular oxidative stress in HO-1, Nrf2, CAT, and SOD; (**B**) WB analysis of HO-1 and Nrf2 in L02 cells; (**C**) assessment of the total ROS levels in L02 cells. The data are expressed as the mean ± SD (*n* = 6). *** *p* < 0.001 vs. the NC and control group; ### *p* < 0.001 vs. the ALC group.

**Figure 6 ijms-23-07701-f006:**
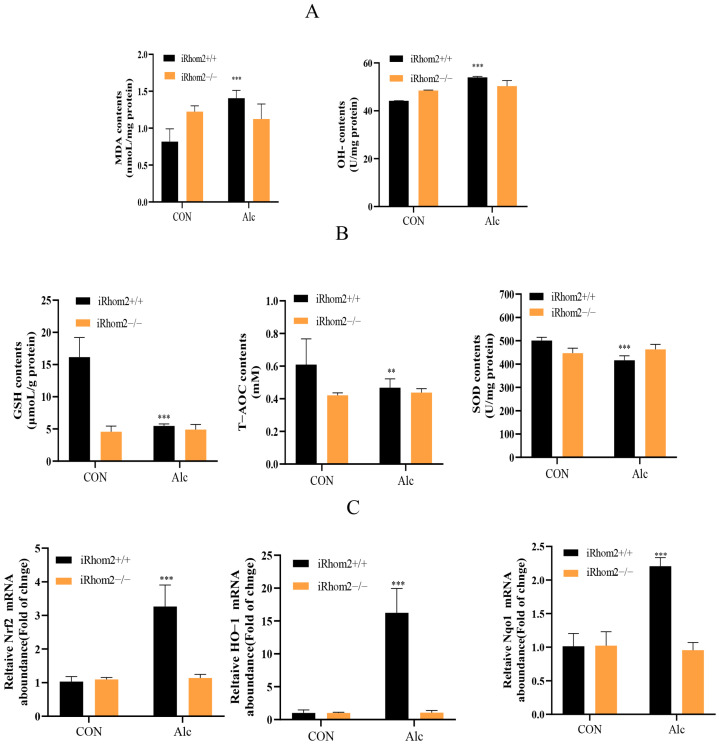
The iRhom2 knockout can reduce oxidative stress and JNK activation in alcohol-induced liver fibrosis in mice; (**A**) liver-oxidative-stress-related indicators, including total MDA and OH-; and (**B**) antioxidant activities of SOD, GSH, and T-AOC; (**C**) HO-1, Nrf2, and Nqo1 in the liver detected with qPCR; (**D**) WB analysis of phosphorylated JNK; (**E**) IF analysis of phosphorylated JNK in liver tissue sections. The data are expressed as an average ± SD (*n* = 6). ** *p* < 0.01, and *** *p* < 0.001 vs. the control group (iRhom2+/+ group).

**Figure 7 ijms-23-07701-f007:**
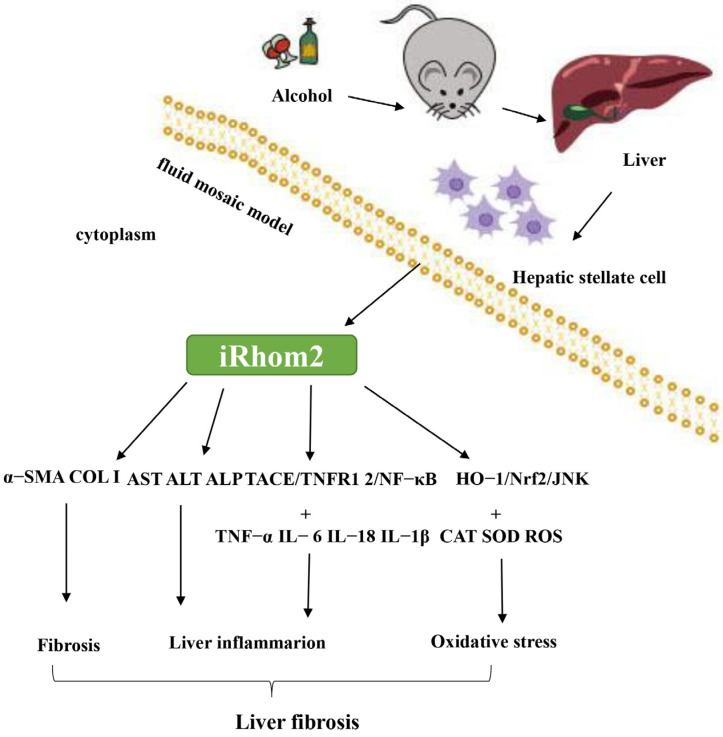
Models showing that iRhom2 is a positive regulator of alcohol-induced liver fibrosis; iRhom2 plays an important role in regulating the progression of liver fibrosis in alcohol-exposed mice, most likely by activating the TACE/TNFRs/NF-κB signaling pathway to promote inflammation. In addition, iRhom2 exacerbates liver fibrotic lesions by activating JNK expression to enhance oxidative stress via the Nrf2/HO-1 pathway.

**Figure 8 ijms-23-07701-f008:**
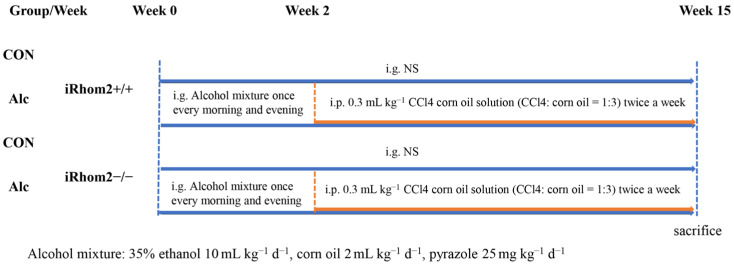
Schematic of the experiment design.

## Data Availability

Data are not publicly archived. Data analyzed and generated during the study are available from the corresponding author Liancai Zhu on reasonable request.

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
