# Peer review of "Deficiency in Inactive Rhomboid Protein2 (iRhom2) Alleviates Alcoholic Liver Fibrosis by Suppressing Inflammation and Oxidative Stress"

_ijms, 2022, doi:10.3390/ijms23147701_

Round 1

Reviewer 1 Report

Dear Authors,

Thank you so much for submit your valuable study to the IJMS. This study was well designed and extract great results and conclusion, however  the current study should have to improve before acceptance. 

-> Please make a brief version of experimental scheme to easily understanding to the readers. 

-> Can authors put the each of specific control gorups? CCl4 only, Pyrazole only?

-> Since the authors mentioned about pyrazole, where did you put the results for the detoxification enzymes against alcohol metabolism in the liver tissue?

-> Please mention about how authors make the KO iRhom KO conditions to the mice? There is no explanations in the manuscript.

Cells and cell culture

Rat hepatic stellate cells (HSC-t6) were purchased from Shanghai Cell Research In stitute (Shanghai, China). Human liver (L02) cells were donated by Professor Liling Tang of Chongqing University.

=> Please mention that why authors did use different host species cell types. And The L02 is adaptable for Alcoholic liver disease model?

- Please provide the protocol or experimental procedures of in vitro model of ALD.

- The WB images and IF images are necessary to change with high qualty.

1. Animal model: The mixture of alcohol (10 mL/kg. D), corn oil (2 mL/kg. D) and  pyrazole (25 mg/kg. D) were given by gavage, once in the morning and once in the even ing. From the second week, CCl4 olive oil solution (CCl4: olive oil=1:3) was injected intra- peritoneally at the dose of 0.3mL/kg twice a week, until the end of the 15th week. 

Reviewer 2 Report

Authors reported that iRhom2 is a key regulator that promotes inflammatory responses and regulates oxidative stress in alcoholic liver fibrosis lesions. iRhom2 is potentially a new therapeutic target for alcoholic liver fibrosis.

1.      What is TACE? Is TACE Tumor necrosis factor-alpha-converting enzyme/ADAM17? Do not use the abbreviations when they first appear in the abstract and text.

2.      In abstract section, what is “TNF-“?

3.      In Introduction section, explain the TACE signaling in more detail.

4.      In line 68, need period between …are in activation” and “The state…”

5.      In Figures 2-6, what are “Con” and “ALC”??

6.      Authors should more explain about alcohol experiment. Ethanol??

7.      In the title, authors should spell out of iRhom2.

8.      Authors should ask native English speaker to edit their manuscript.

Round 2

Reviewer 1 Report

Thank you  so much for your effort to response reviewer's comments.

The current version is available to accept!

Reviewer 2 Report

All queries are addressed.